# Spherical Planting Inversion of GRAIL Data

Guangyin Lu [1], Dongxing Zhang [1], Shujin Cao [1,2,3,*], Yihuai Deng [2], Gang Xu [1], Yihu Liu [1], Ziqiang Zhu [1] and Peng Chen [2]

[1] School of Geosciences and Info-Physics, Central South University, Changsha 410083, China
[2] School of Earth Sciences and Spatial Information Engineering, Hunan University of Science and Technology, Xiangtan 411201, China
[3] Institute of Geophysics & Geomatics, China University of Geosciences, Wuhan 430074, China
* Correspondence: shujin.cao@hnust.edu.cn

**Abstract:** In large-scale potential field data inversion, constructing the kernel matrix is a time-consuming problem with large memory requirements. Therefore, a spherical planting inversion of Gravity Recovery and Interior Laboratory (GRAIL) data is proposed using the L1-norm in conjunction with tesseroids. Spherical planting inversion, however, is strongly dependent on the correct seeds' density contrast, location, and number; otherwise, it can cause mutual intrusion of anomalous sources produced by different seeds. Hence, a weighting function was introduced to limit the influence area of the seeds for yielding robust solutions; moreover, it is challenging to set customized parameters for each seed, especially for the large number of seeds used or complex gravity anomalies data. Hence, we employed the "shape-of-anomaly" data-misfit function in conjunction with a new seed weighting function to improve the spherical planting inversion. The proposed seed weighting function is constructed based on the covariance matrix for given gravity data and can avoid manually setting customized parameters for each seed. The results of synthetic tests and field data show that spherical planting inversion requires less computer memory than traditional inversion. Furthermore, the proposed seed weighting function can effectively limit the seed influence area. The result of spherical planting inversion indicates that the crustal thickness of Mare Crisium is about 0 km because the Crisium impact may have removed all crust from parts of the basin.

**Keywords:** spherical planting inversion; tesseroids; GRAIL data; Mare Crisium; seed weighting function

## 1. Introduction

Kernel matrices are usually generated using partial differential and integral equation methods in gravity field inversion. Partial differential equations, such as the finite-difference [1,2], finite-element [3], and finite-volume methods [4,5], are used to discretize Poisson's equation to obtain potential fields. However, these methods are difficult to apply to gravity inversion because higher-order derivatives involve higher-order differences or the fast Fourier transform. Comparatively, integral equations utilize the superposition principle to calculate the gravity contribution of complex geological bodies.

The observation surface of gravity data in small areas can be approximated as a plane; in the Cartesian coordinate system, the potential field calculation in the spatial domain is usually performed using prisms or general polyhedral cells [6–9]. In large regional and even global gravity problems, the observation surface is generally expressed and processed in spherical coordinates because of the need to consider the observation surface's curvature [10]. As a result, interpretation models are usually divided into tesseroids in a spherical coordinate system using the parameters radius, latitude, and longitude. The gravitational effect generated by each unit is calculated and then summed to obtain global gravity/tensor gravity data.

As a result of the CHAMP (Challenging Mini-Satellite Payload), GRACE (Gravity Recovery and Climate Experiment), GOCE (Gravity Field and Steady-State Ocean Circula-

tion Explorer), and GRACE Follow-On satellite launches [11], it is now possible to obtain large-scale high-precision gravity data economically.

Inversion of the gravity field is a powerful tool for understanding the density distribution inside planets. Many gravity inversions have been proposed in recent years. Liang et al. applied tesseroid units to the lunar GRAIL gravity inversion and obtained the 3D density structure of the Moon [12]. Uieda et al. [13] analyzed the Moho surface depth of the South American continent using tesseroid grids for a fast nonlinear inversion of GOCO5S model data in a spherical coordinate system. Zhang et al. [14] developed an efficient algorithm for gravity gradient full-tensor data, which was used to study the density distribution of the Mare Smythii mascon on the Moon. Zhao et al. [15] performed a 3D spherical gravity inversion with tesseroid units for the GL1500E model data to obtain the 3D density distribution of the Imbrium and Serenitatis mascon basins.

ETOPO1 is a one-arc-minute global relief model of the Earth's surface that integrates land topography and ocean bathymetry provided by the National Oceanic and Atmospheric Administration [16]. Take a spherical gravity inversion with the ETOPO1 model as an example. When a 1- by 1- minute latitude-longitude grid is used to cover the globe, the number of observation points is $180 \times 60 \times 360 \times 60 = 2.3328 \times 10^8$. The number of kernel matrix elements for potential inversion is the product of the number of tesseroid units and observation points. The memory requirements of the kernel matrix, whether storing double-precision or single-precision data, are far more than the physical memory size of existing mainstream workstations.

Constructing kernel matrices is a time-consuming problem with high memory requirements for potential inversion, especially for large-potential data [17]. Fast and accurate three-dimensional gravity inversion of large-scale, multi-scale and massive data is currently performed by improving the computational performance of equipment, reducing data dimensionality, and speeding up the convergence of the inversion iterations.

First, high-performance computation relies on multi-core CPUs, clustered computer systems, and GPUs. Then, massively parallel acceleration studies are performed to reduce time-consuming forward modeling and inversion [18]. For example, Moorkamp et al. [19] and Chen et al. [20] used massively parallel computing techniques to achieve rapid forward modeling of gravity and gravity gradient components. However, since such calculations do not store the kernel matrix, each inversion iteration in practical applications involves several forward calculations, resulting in relatively low efficiency. In addition, there are only a few published papers because of its relatively high entry barriers in terms of hardware and software.

Second, using constraint functions to accelerate gravity inversion convergence, such as optimizing recovered density distributions by a depth-weighting function [21], ensures fast convergence of the objective function by applying a preconditioning matrix [22], and adding a density constraint to optimize the upper and lower bounds of the inversion results [23,24]. Alternatively, inversion may be constrained by using seismic [25], magnetotelluric [26], and other geophysical investigations [27]. Various filters are also used to suppress or eliminate coherent noise [28,29].

Finally, for reducing the dimensionality of the gravity inverse problem, numerous methods such as the wavelet transform [30], footprint inversion [31], and fast Fourier transform [32] are used to reduce the memory requirements of kernel matrices and significantly improve computational efficiency. Taking wavelet transform techniques as an example, Li and Oldenburg [33] represented the kernel coefficient matrix as a sparse matrix based on the wavelet compression technique, effectively decreasing the number of nonzero matrix elements. However, it loses signal accuracy to some extent. René [34] proposed an open, reject and fill (Open–Reject–Fill) iterative criterion for the gravity inversion method, which can grow one element at each inversion iteration. However, this method relies heavily on geological priori information. Using a systematic search algorithm, Camacho et al. [35] selected optimal prisms that minimized the objective function at each iteration

until the termination criteria were met. Finally, this yields a three-dimensional residual density distribution.

Following René's method [34], Uieda and Barbosa [36] proposed a "grown" gravity inversion with the total variation regularization function given by Silva Dias et al. [37] to avoid calculating and storing kernel matrices. The algorithm grows a prism around a specific prism (called a "seed") to obtain the boundary information of tectonics. Nevertheless, inversion results heavily depend on the correct location, density, and seed number. Thus, Uieda and Valeria [37] achieved a robust planting inversion based on a shape-of-anomaly misfit function, an L2-norm data-misfit function, and a horizontal weighting function. However, weighting functions are challenging for inversions with various seeds, especially for complex anomalies.

This paper introduces a spherical planting inversion into a spherical coordinate system. First, we develop the principles of spherical planting inversion for GRAIL data and further improve the model-misfit function by using a modified distance function. Subsequently, a robust adjustment is applied to the inversion process using a novel seed weighting function based on the covariance matrix for the given data. Then, a spherical planting inversion is performed with a composite model from Liang et al. [12] to verify its correctness and efficiency. Mare Crisium is chosen as a study area, and the residual Bouguer gravity data is derived from the lunar gravity model GRGM1200A [38]. We use Parker–Oldenburg's algorithm [39], a radially averaged energy spectrum, and Wieczorek's (2013) algorithm [40] to determine the locations and density contrast of the seeds. The spherical planting inversion with the residual Bouguer gravity data is performed to obtain 3-D density distributions of Mare Crisium. Finally, we discuss the inverted density anomalies of the spherical planting inversion.

## 2. Materials and Methods

### 2.1. Spherical Planting Inversion

Following the ideas of Uieda and Barbosa [36,37], to avoid solving an extensive system of linear equations constructed by density imaging, the spherical planting inversion of GRAIL data is proposed using an L1-norm-based systematic search algorithm to reduce memory requirements and facilitate fast convergence by replacing iterative inversion with cumulative summation analysis. For this reason, a residual vector $r$ between the observation data $d$ and the prediction data $g$ is constructed by the L1-norm based on Equation (1).

$$r = d - g \tag{1}$$

The data-misfit function is defined (based on the L2-norm) by least squares estimation, which can be expressed as

$$\phi_d = \|r\|_2 = \left( \sum_{i=1}^{N_d} (g_i - d_i)^2 \right)^{1/2} \tag{2}$$

where the subscript $i$ corresponds to the $i$-th observation point. $N_d = n_x \times n_y$. $n_x$ $n_y$, and $n_r$ represent the section number of the interpretation model along the $x$-, $y$- and radial axes, respectively.

Compared to the L2-norm, the L1-norm is not sensitive to data with significant errors so it can provide a robust estimation strategy.

$$\phi_d = \|r\|_1 = \sum_{i=1}^{N_d} |g_i - d_i| \tag{3}$$

Conventional inversion applies regularization terms to reduce the instability and non-uniqueness of the solutions. Many inverse strategies have been proposed for recovering compact anomalous sources, such as maximizing their compactness [41], concentrating

them along one or more axes [42,43], and minimizing their area/volume [44]. For example, René [34] proposed a systematic search inversion of gravity data using the "shape-of-anomaly" function to reconstruct 2D compact bodies (i.e., with no hollows inside). However, this inversion algorithm only uses a single sign/density contrast to recover density distributions. Compared with René's (1986) method, Uieda and Barbosa interpret multiple geologic sources with density contrasts of different signs [36].

Consider that recovering a density distribution from GRAIL data is an ill-conditioned problem, it can be converted into a well-conditioned problem by adopting the following constraints:

(1) A compact scheme ensures that there are no voids in the recovered density distributions, which are obtained from spherical gravity inversion with a seed [36,41–43].

(2) An optimal prism is selected one by one from a neighboring tesseroids list, which is derived from a seed with density contrast $\rho_s$, and shares faces with the seed or its adjacent tesseroids. Then, the optimal prism's density value $\rho_j$ is set to the same as the seed.

(3) For other uninvolved tesseroids, their density value equals zero. Based on the above constraint strategies, the objective function of the spherical planting inversion of GRAIL data is given by

$$P^a(\boldsymbol{m}) = \phi_d + \mu\phi_m \rightarrow \min \tag{4}$$

where, the regularization parameter $\mu$ balances the trade-off between the data-misfit function $\phi_d$ and the model-misfit function $\phi_m$, and is defined in the model space that imposes geologic constraints on the recovered density distribution.

$$\phi_m = \sum_{k=1}^{N_s} \left( \sum_{j=1}^{N_m} \frac{\rho_j}{\rho_j + \varepsilon} \varsigma(j,k) l_{j,k}^\beta \right) \tag{5}$$

where, $\rho_j$ is the $j$-th element of the density vector $\boldsymbol{m}$, whose size is $N_m$, and $N_s$ is the number of seeds. $\varepsilon$ is a small positive count and will avoid a singularity error when $\rho_j = 0$. The sign function $\varsigma(j,k)$ returns 1 when the $j$-th tesseroid is derived from the $k$-th seed, as shown in Figure 1; otherwise, it returns 0. When $N_s = 1$, Formula (5) is similar to Formula (10) in Uieda and Barbosa's paper [36]. $l_{j,k}^\beta$ is the distance from the $j$-th tesseroid's centroid to the $k$-th seed's centroid.

Regardless of how a tesseroid's length and width vary with its position, the modified version $l_{j,k}^\beta$ is given by

$$l_{j,k}^\beta = \sqrt{(j_x - k_x)^2 + (j_y - k_y)^2 + (j_r - k_r)^2} \tag{6}$$

where $(j_x, j_y, j_r)$ and $(k_x, k_y, k_r)$ stand for the order of the $j$-th tesseroid and the $k$-th seed along the longitude, latitude, and radius, respectively. Here, $j = j_x + (j_y - 1) \times n_x + (y_r - 1) \times n_x \times n_y$.

The solution of the spherical planting inversion of GRAIL data is recovered through a growth procedure divided into the following four steps, as shown in Figure 1.

1. Initially, this algorithm requires a set of $N_s$ seeds, each of which is a tesseroid of the geophysical model. Seed locations can be obtained by Euler deconvolution or by manual selection, and the user can specify the density contrast of seeds. Meanwhile, the density contrast of all other tesseroids equals zero.

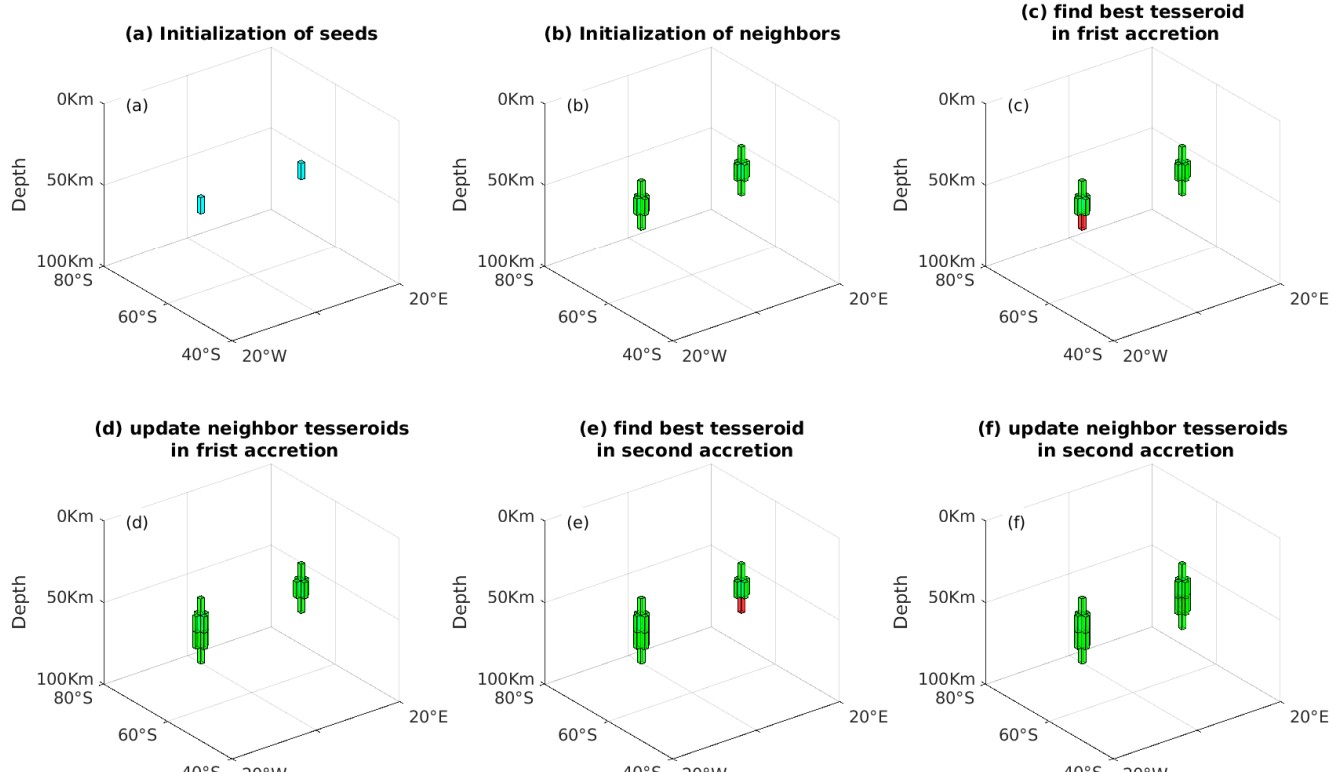

**Figure 1.** Four stages of the spherical planting algorithm: (**a**) Initialization of seeds (in cyan) at the first stage; (**b**) Initialization of neighboring tesseroids (in green) of seeds at the second stage; (**c**) Finding the optimal tesseroid (in red) at first accretion, and (**d**) updating neighboring tesseroids of the first seed at the third stage; (**e**) Finding the optimal tesseroid at second accretion, and (**f**) updating neighboring tesseroids of the second seed at the fourth stage.

2. Secondly, for each seed, a list is constructed of neighboring tesseroids that share a face with the seed or neighboring ones. In constraint 3, only seeds will have a specific density contrast, and all other tesseroids' density contrasts are set to zero. Then the initial total objective function $P^{(0)}$ is given by

$$P^{(0)} = \phi_d^{(0)} + \mu\phi_m^{(0)} \tag{7}$$

where, the initial total data-misfit function is given by

$$\phi_d^{(0)} = d - \sum_{i=1}^{N_s} \rho_s a_s \tag{8}$$

Here, $\rho_s$ is the density-contrast value of the *s*-th seed, and the vector $a_s$ consists of kernel functions which are gravitational effects at all observation points due to the *s*-th seed. Due to there being no neighboring tesseroids in the initial step, $\phi_m^{(0)} = 0$.

3. An iteration involves *Ns* accretions in a spherical planting inversion. This algorithm finds a tesseroid from its neighboring tesseroid list for every accretion by minimizing the objective function and guaranteeing convergence of the following two conditions:

$$\left|\phi_d^{(new)} - \phi_d^{(old)}\right|/\phi_d^{(old)} \geq \delta \tag{9}$$

where the total data-misfit function $\phi_d^{(new)}$ is calculated for the chosen neighboring tesseroids in the current accretion and all tesseroids whose density contrast is not equal to zero, $\phi_d^{(old)}$

is the total data-misfit function at the previous accretion. The small positive scalar $\delta$ ranges from $10^{-2}$ to $10^{-6}$.

$$\left| P^{(new)} - P^{(old)} \right| / P^{(old)} \geq \tau \tag{10}$$

where $P^{(new)}$ and $P^{(old)}$ are total global functions of current accretion and precious accretion, respectively.

Ensuring the chosen tesseroid satisfies Equations (9) and (10), and treating it as the optimal tesseroid in the current iteration, then, the algorithm changes its density contrast from zero to the seed's and removes it from the seed's neighboring tesseroids list to guarantee constraint 3; otherwise, the algorithm will traverse over other seeds.

4. In any accretion at the previous iteration, if the convergence conditions (Equations (9) and (10)) are met, the growth process continues. Rather, the spherical planting inversion should be terminated, and a 3D density distribution should be obtained.

To avoid the recovery density distribution of spherical planting inversion being strongly influenced by the parameters of seeds, Uieda et al. [37] proposed a "shape-of-anomaly" data misfit that measures the difference in shape between the observed and predicted data and is insensitive to differences in amplitude. By introducing "shape-of-anomaly" data misfit, the data-misfit function is defined as:

$$\phi_d = \sqrt{\sum_{i=1}^{N_d} (a\boldsymbol{g}_i - \boldsymbol{d}_i)^2} \tag{11}$$

where $\alpha$ is the scaling factor; for a given predicted data $\boldsymbol{g}$, $a$ can be calculated by transforming the above equation:

$$a = \sum_{i=1}^{N_d} (\boldsymbol{g}_i \boldsymbol{d}_i) / \sum_{i=1}^{N_d} (\boldsymbol{d}_i)^2 \tag{12}$$

In the spherical planting inversion, the reconstructed density distributions are determined by the seeds' parameters. As the reconstruction model's size increases, the number of neighboring tesseroids gradually increases, and the distance between seeds and neighboring tesseroids increases. This causes neighboring tesseroids in a particular direction always to be determined as optimal tesseroids, eventually leading to mutual intrusion between adjacent anomalous sources. This issue is addressed by the weighting function, which limits the seeds' influence area and makes them ignore far abnormal sources.

$$w_h = \exp\left( \frac{(x_s - x)^2 + (y_s - y)^2}{(r_s)^2} \right)^{-\beta_h} \tag{13}$$

where $(x_s, y_s)$ and $(x, y)$ are the coordinates of the $s$-th seed and observation point, respectively, and $r_s$ is the influence radius of the $s$-th source. For the inversion of gravity data, the horizontal decay factor $\beta_h$ ranges from 1 to 3. Then, the data-misfit function (11) can be rewritten as:

$$\phi_d = \sqrt{(aw_h\boldsymbol{g} - \boldsymbol{d})^{\mathrm{T}}(a\boldsymbol{g} - \boldsymbol{d})} \tag{14}$$

where T is the transpose operator.

However, it is challenging to choose the right values for Equation (13), particularly when the quantity of seeds is large or the gravity anomaly is complicated. To overcome this difficulty, by referring to the data covalence matrix $\boldsymbol{W}_d$ in the density image, we can define a novel seed weighting function:

$$\overline{w}_h = diag(\boldsymbol{W}_d) = diag(\{1/\sigma_1, \cdots, 1/\sigma_n\}) \tag{15}$$

where $\sigma_i$ is the error standard deviation associated with the $i$-th observation data.

The process of the spherical planting inversion of GRAIL data is detailed in a flow chart, shown in Figure 2.

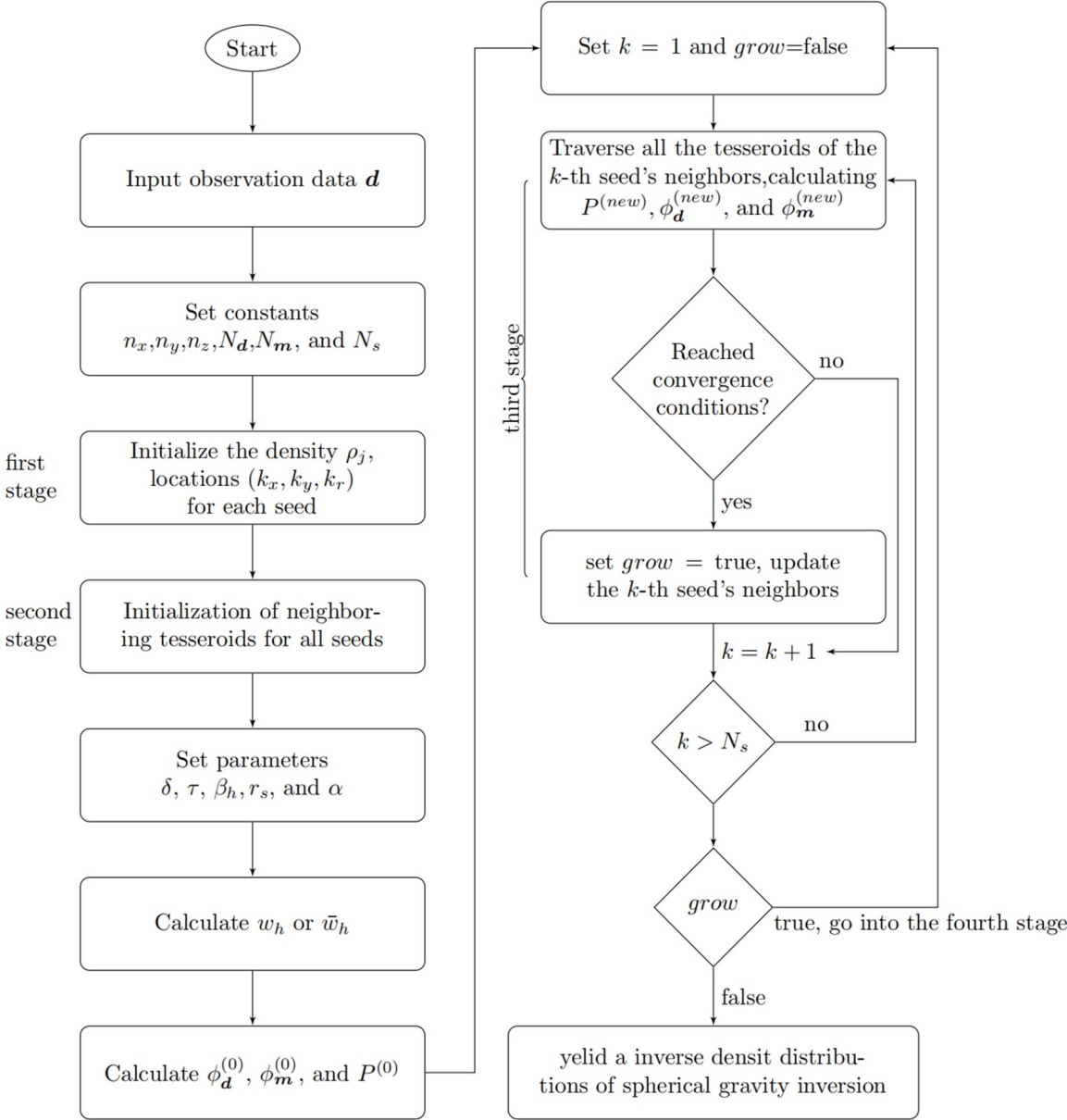

**Figure 2.** Flow chart for the spherical planting inversion.

*2.2. Model Studies*

2.2.1. Verification of the Validity of the Novel Seed Weighting Function $\overline{w}_h$

First, a simple synthetic model with an isolated anomalous source was proposed by Zhang et al. [14] and is used as an example to compare the reconstruction effect of the two weighting functions mentioned in Equations (13) and (15), respectively. The isolated anomalous source is in the blue frame as in Figure 3 and with 0.5 g/cm³ density contrast. Then, a rectangular survey grid, with 40 points, with a 0.5° grid interval along the longitude and latitude directions, is employed to obtain gravity data. A seed with locations 35°, 35°, and 1673 km along latitude, longitude, and radius directions, respectively, and with a density contrast of 0.5 g/cm³, is employed for performing the spherical gravity inversion for gravity data, which is contaminated by 3% Gaussian noise of the gravity data.

Using a seed with locations at 35°, 35°, and 1673 Km along latitude, longitude, and radius directions, respectively, and with a density contrast of 0.5 g/cm³, we perform spherical gravity inversion for gravity data, which is contaminated by 3% Gaussian noise in the forward result.

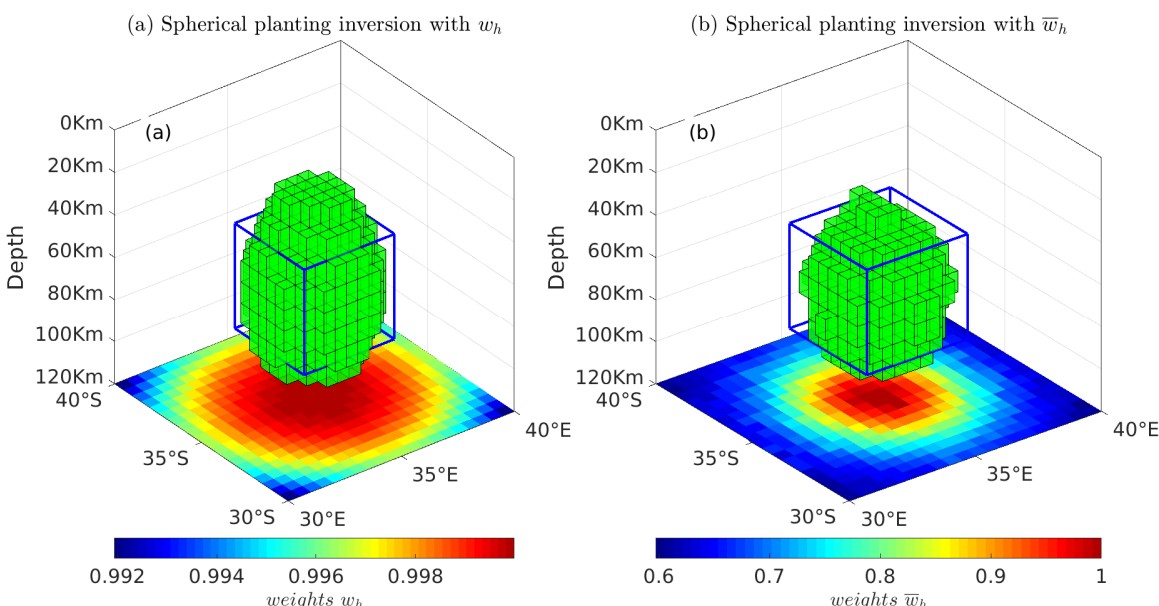

**Figure 3.** Recovered density distribution of spherical planting inversion of $g_z$ data. (**a**) With the horizontal weighting function $w_h$. (**b**) With the novel seed weighting function $\overline{w}_h$. The weight distributions as shown in the $z = 120$ km plane. $r_s = 50$, $\beta_h = 1.2$, $\delta = 0.05$, and $\tau = 5.0 \times 10^{-4}$.

Regarding relatively simple anomalous sources, it is feasible to set a horizontal weighting function for multiple seeds, which can effectively limit the seed's influence area and avoid the intrusion of the estimation grid of adjacent seeds into each other. Despite this, as far as the characteristics of the horizontal weighting function $w_h$ are concerned, its influence range is a circle. Consequently, it is too expensive to artificially set the horizontal weighting function for multiple seeds, especially for complex gravity data. The recovered density distributions in Figure 3a are far beyond the actual model in the depth direction, which is caused by the gravitational contribution of the tesseroid becoming smaller with the increase in depth. Meanwhile, the corresponding data misfit function becomes smaller, so the optimal tesseroid will gradually increase along the depth direction. Eventually, the reconstructed density model will be far beyond the actual model in the depth direction. For this reason, drawing on the role of the data weighting function $W_d$ in density imaging, we construct a novel seed weighting function. Compared to the recovered density distribution derived from $w_h$ as shown in Figure 3a, the spherical planting inversion with the horizontal weighting function yields a more compact result in Figure 3b.

### 2.2.2. Verification of the Validity of Spherical Planting Inversion

As shown in Figure 4, we verify the proposed algorithm using the composite model following Zhang et al. [14] with two tesseroids at the same depth for easy comparison. One model occupied the volume from 32° to 34° longitude, 34° to 36° north latitude, and 1678 to 1718 km in depth, with a density contrast of 0.2 g/cm³. The other model had a density contrast of 0.7 g/cm³, and its geological setting ranged from 36° to 38° longitude, 34° to 36° latitude, and 1658 to 1698 km in depth. A total of 62,500 data points is calculated 10 km above the lunar surface with an interval of 0.8° × 0.8° along latitude and longitude. The field source space is divided into 20 × 20 × 10 = 4000 tesseroids, and each tesseroid is 0.8° × 0.8° × 10 km along latitude and longitude, and radius direction, respectively. Figure 5 shows the data contaminated with pseudo-random Gaussian noise with zero means and five mGal standard deviations. Then, we invert the contaminated data with two seeds; one seed is located at [33°, 35°, 1698 km] with 0.2 g/cm³ density contrast, and the coordinates of the other seed were 35°, 37° and 1678 km along latitude, longitude and radius direction, respectively, with a density contrast 0.7 g/cm³.

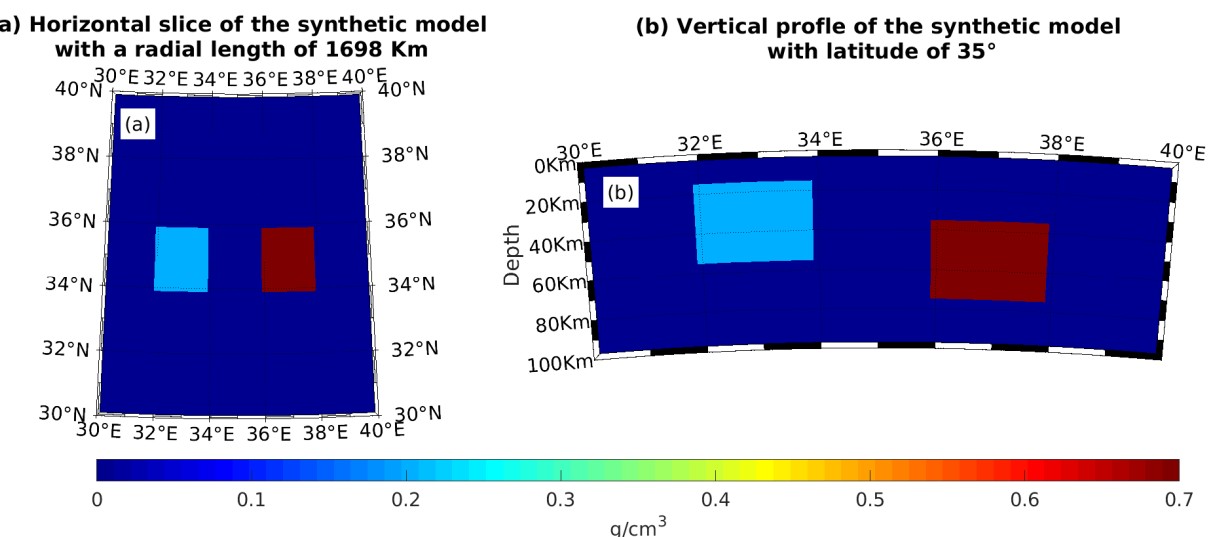

**Figure 4.** Composite model: (**a**) A horizontal slice of the composite model with a radial length of 1698 km. (**b**) A vertical profile of the composite model with a latitude of 35°. $\delta = 2.0 \times 10^{-5}$ and $\tau = 1.0 \times 10^{-6}$.

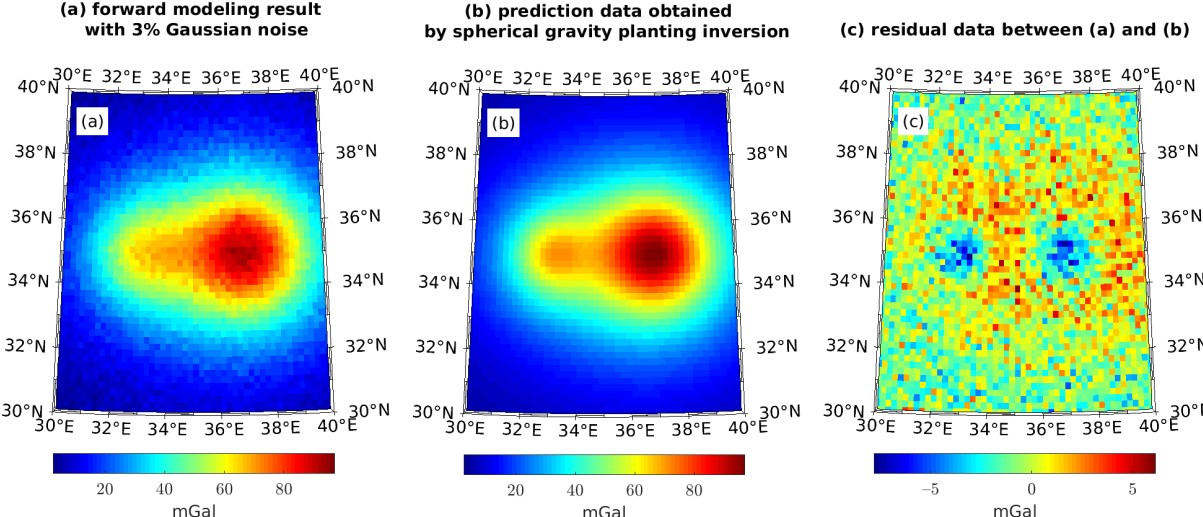

**Figure 5.** Gravity maps: (**a**) Forward modeling result. (**b**) Prediction data obtained by spherical planting inversion. (**c**) Residual data between forward modeling result (**a**) and prediction data (**b**). $\delta = 0.05$, and $\tau = 5.0 \times 10^{-4}$.

It is difficult for the Occam inversion to recover depth resolution from surface gravity data because there is no depth resolution in the data [21,45,46]. The interfaces of the two models in the spherical Occam inversion results are ambiguous, even though the depth weighting matrix is used. In contrast, the density slices of the spherical planting inversion are consistent with those of the forward model in Figure 6, which verifies the correctness of the algorithm presented in this paper.

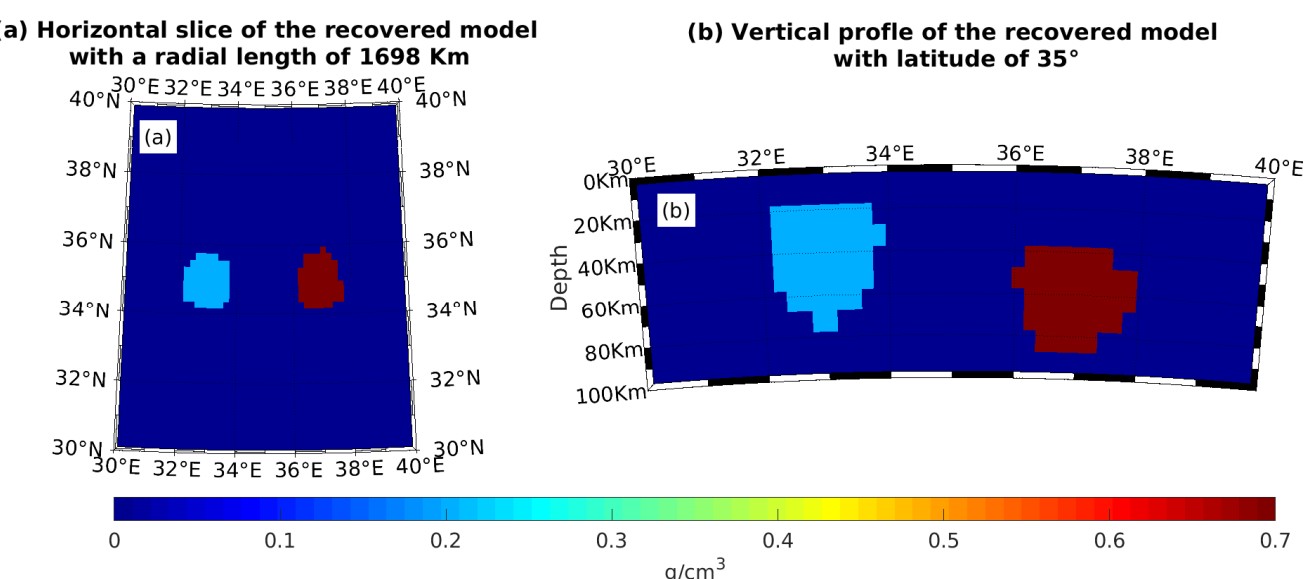

**Figure 6.** Density-contrast distribution maps. (**a**) Horizontal slice of the reconstruction model along with a radial length of 1698 km, and (**b**) vertical profile of the composite model along a latitude of 35°.

## 3. Results

The discovery of lunar mascons was one of the most significant results of early gravity field studies in lunar exploration [47,48]. Moreover, Mare Crisium is one of the largest mascons on the Moon [49] and is an impact-formed poly-cyclic basin [50,51] that formed during the Nectarian Period [52,53], located between 10° and 30° N and 50° and 70° E, in the northeastern part of the lunar front (centered near 17.0° N and 58.8° E), with a basin diameter of 555 km and covering an area of about 17,600 km$^2$. The main ring of the crater rim is about 740 km in diameter and is one of the deepest basins on the lunar surface (4.57 km). Therefore, we selected Mare Crisium as an example to elucidate the mechanism of spherical planting inversion.

### 3.1. Data

The gravity model GRGM1200A, applied in Bouguer correction by NASA, expanded to degree and order 1200 with sensitivity down to <5 km resolution from the GRAIL mission, is used in this paper [54]. Deliberating the very high correlation between gravity and topography at L = 7~700 [55], following Deng et al. [56], the spatial resolution of the gravity field is related to the spherical harmonic coefficient and defined by the half wavelength in spherical inversion [57]. When the grid size of the interpretation model is 0.8° × 0.8° on a spherical earth, according to the relation between half wavelength lambda and the suitable degree *N* of the gravity model in Liang et al. [12], the suitable degree *N* = 450 [58]. Therefore, in Figure 7, the residual Bouguer gravity is expanded between spherical degrees 7 and 450 [57] to suppress the influence of noise from the high degrees and filter out signals from the deep interior (such as the crust–mantle boundary or Moho) from the low degrees for highlighting lunar mascons [59–61]. The topography of the Crisium mascon is derived from the spherical harmonic topography model MoonTopo2600p, as shown in Figure 8. Then, 20 × 20 = 400 observation data points are calculated at 10 km height relative to the reference radius of 1738 km using the gravity field model mentioned above, as shown in Figure 9.

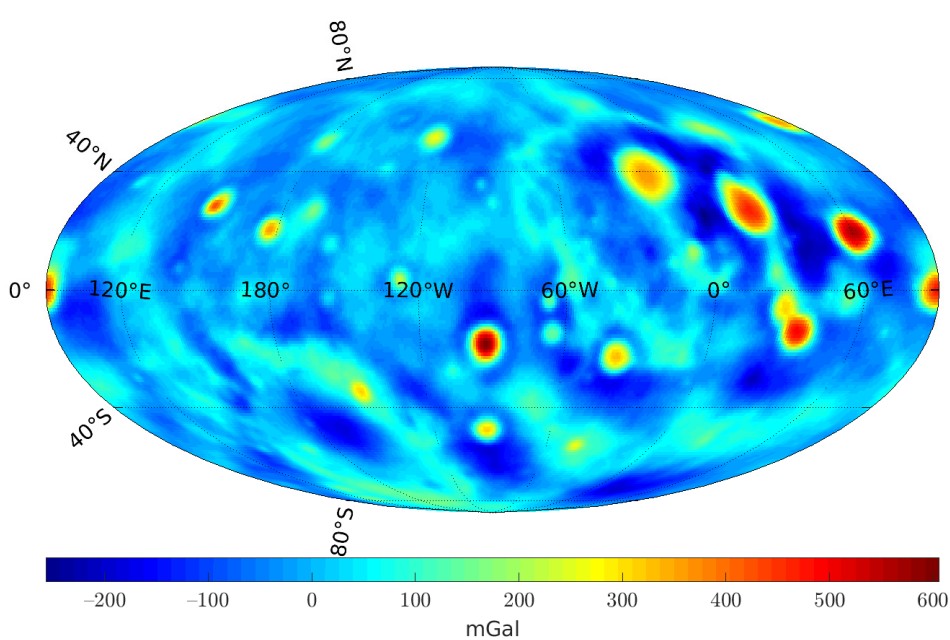

**Figure 7.** A filtered Bouguer map (l = 7−450) highlights the lunar mascons.

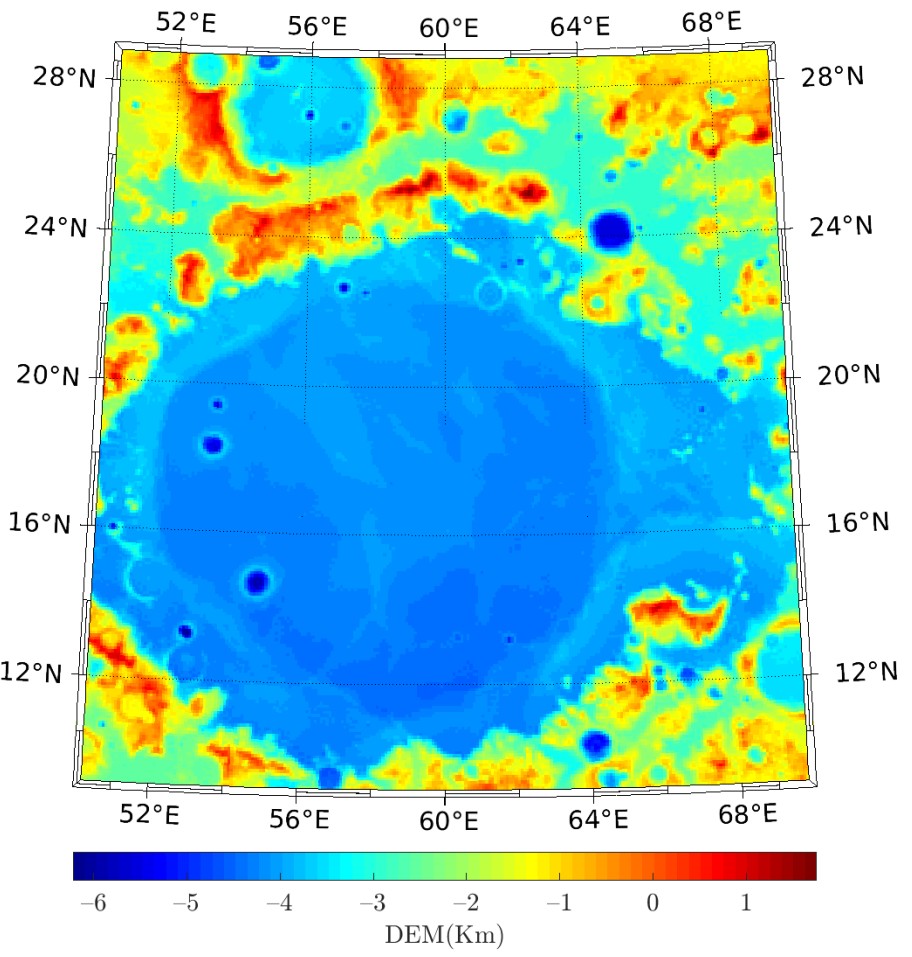

**Figure 8.** The topography of the Crisium mascon.

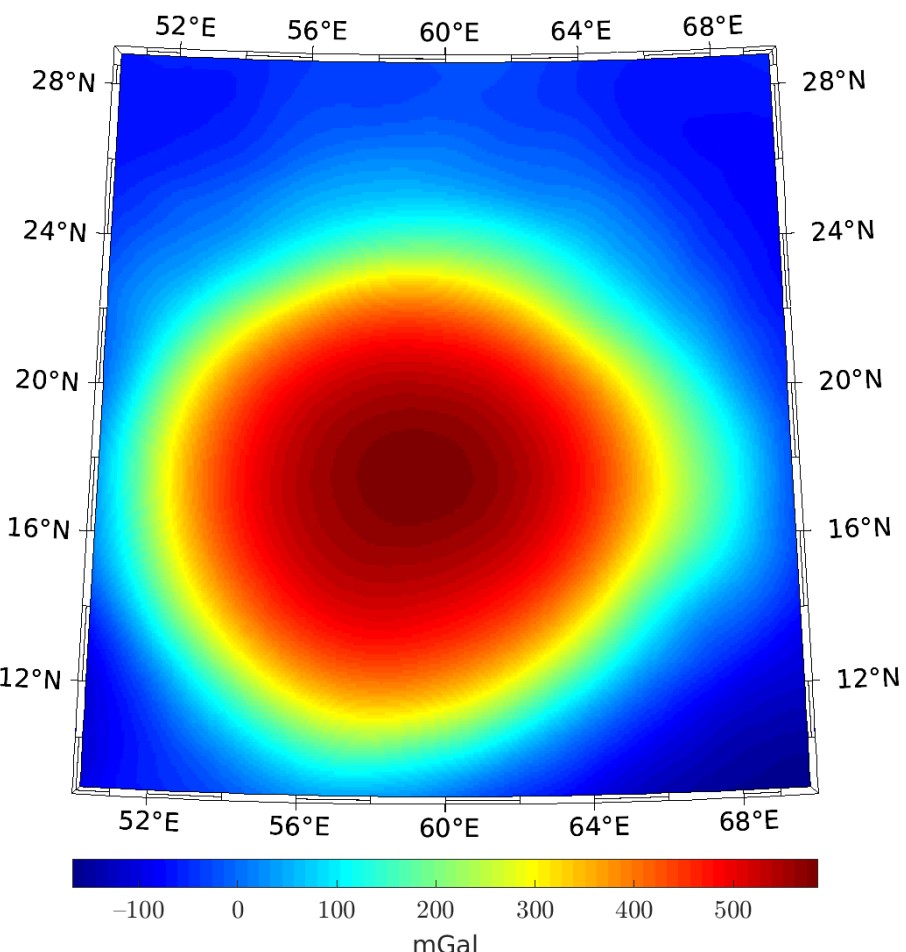

**Figure 9.** The residual Bouguer gravity data.

In Figure 8, Mare Crisium is surrounded by an upland massif [53], with much of its interior and a small part of its marginal upland filled with basaltic rocks and a ring of folded ridges on its outer boundary [62,63]. A folded ridge is a low, linear, or arcuate broad ridge, usually tens of kilometers long. It is placed symmetrically/asymmetrically on an arch close to the circular Maria edge [64,65].

### 3.2. Methodology

A new Bouguer gravity data covering 4°–34° N and 50°–70° E is gridded at an interval of 2 km with a Mercator projection centered at (180 E, 0 N) to determine the depth and density contrast of the Mare Crisium mascons. This area is designed to eliminate the effect of distorted shapes due to the Mercator projection. Assuming that the density contrast is 0.65 g/cm$^3$, the convergence criterion is 0.025 km, the mean crustal reference depth is 30 km [40], the truncation window data length is 2.5%, and the smaller and greater cut-off frequencies are 1/100 and 1/83.3, Parker–Oldenburg's algorithm [39] was employed to invert the residual Bouguer gravity anomaly to reconstruct the crust thickness in this study area. Figure 10 shows that the crust thickness ranges from 0 to 45 km, and the topography variation outlines the impact crater basins.

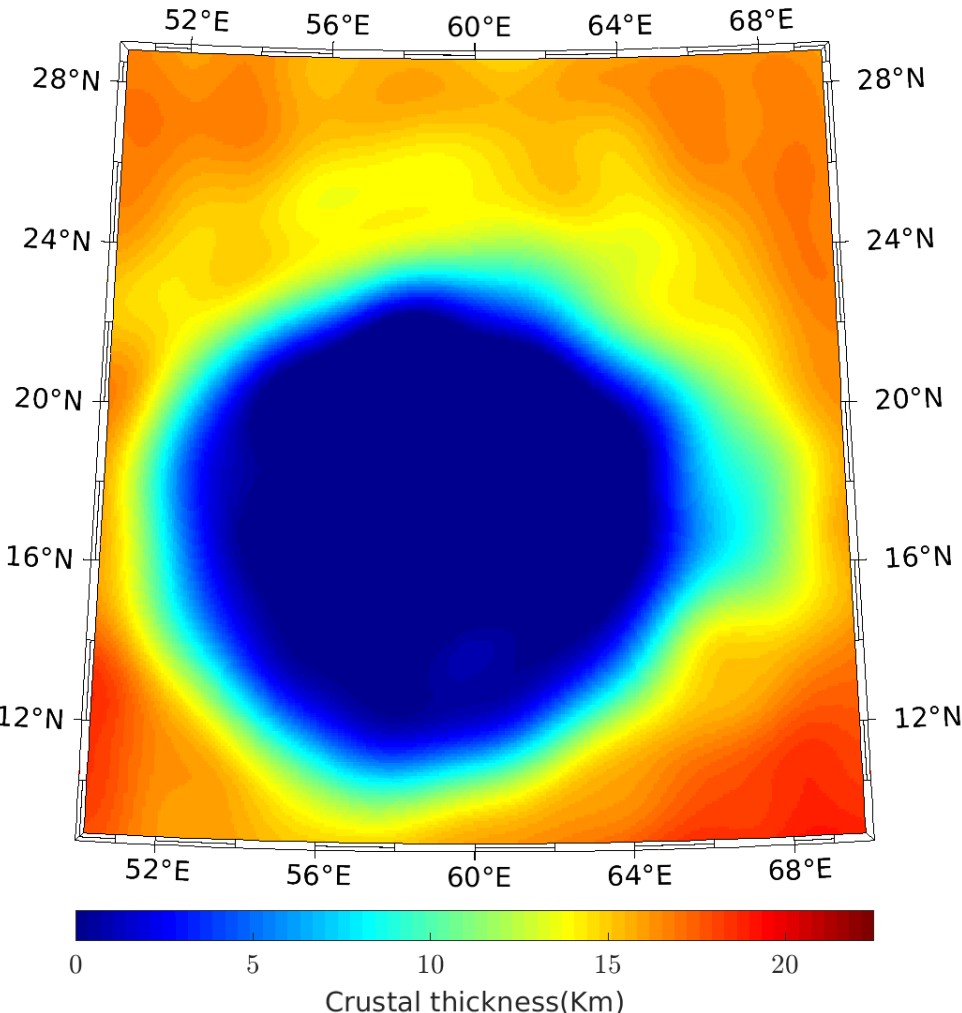

**Figure 10.** The crustal thickness was obtained from the residual Bouguer gravity (Figure 9) using Parker–Oldenburg's method.

The power spectrum method performs regional and residual separations based on a fast Fourier transform for outlining subsurface structures. As an extension of the power spectrum method, the radially averaged energy spectrum yields a typical energy spectrum plot containing three parts corresponding to deep source, superficial source, and noise contribution. Hence, the power spectrum method can also be employed to determine the depths of shallow and deeper structures [66–69].

By following [66,70,71], the radially averaged power spectrum according to frequencies was calculated and is displayed with blue dots in Figure 11. Then, the slopes of the portions can be employed to estimate the depths of anomalous sources. The depth (*z*) for each source corresponding to each segment was calculated by presenting the slope of this part:

$$Z = \frac{-D}{4\pi}$$

where the slope of the dashed line *D* is the ratio between the logarithmic power spectrum and the wavenumber.

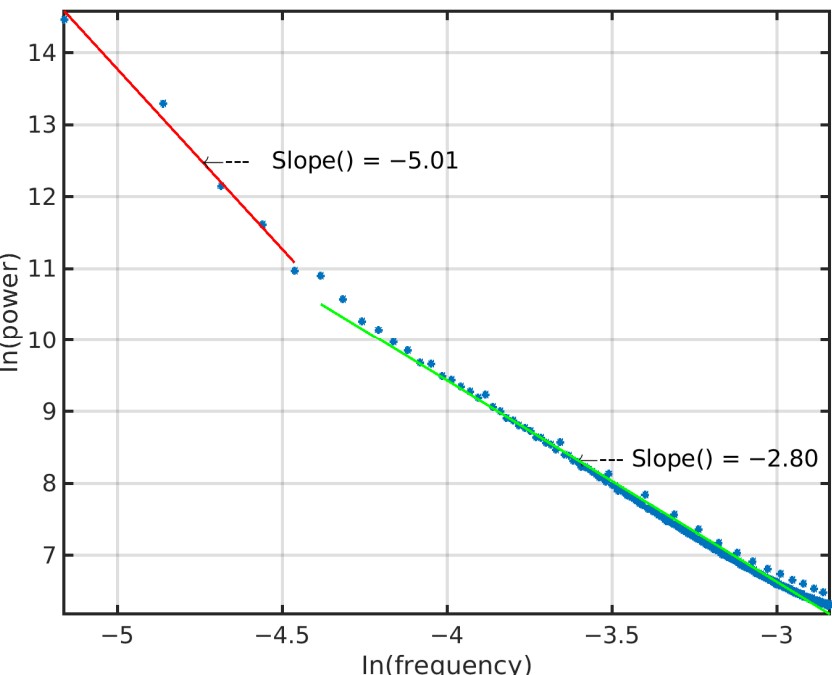

**Figure 11.** Radially averaged power spectrum.

The radially averaged power spectrum curve can be interpreted as two linear slope segments and has been applied to estimate the depth of the gravity interface at 0.4 km, representing the average depth to the surface of the mascon (Figure 11).

In order to further confirm the depth and density of the mascon in this study area, the spherical harmonic density model density_no_mare_n3000_f3050_719.sh is proposed by Wieczorek [72], and the spherical harmonic topography model MoonTopo2600p [73] and the spherical harmonic gravity model GRGM900C [55] are employed to calculate the crustal density, topography, and gravity anomaly, respectively. The crust is assumed to have an average thickness of 35 km, a porosity of 0.12, and a density of 3.22 g/cm$^3$. By following [40] and [74], we used homogeneous density to minimize the correlation between surface topography and Bouguer gravity and determine the crustal thickness. From the Figures 8–12, the crustal thickness in this study area is described as 5 to 40 km. When combined with the topographic map, we can find that the crustal thickness in the middle of this study area is 0.

### 3.3. Spherical Planting Inversion of Mare Crisium

To achieve the spherical planting inversion of residual Bouguer gravity at the Crisium mascon, the geophysical interpretation model covers an area of 9°–29°, 50°–70° and 1638–1738 km in latitude and longitude as well as radial directions. The field source space is divided into 20 × 20 × 10 = 4000 cells, and each cell is 0.8° × 0.8° × 10 km along latitude and longitude, and radius directions, respectively.

Because of the low gravity at each corner and the similar linear background field in Figure 9, following Uieda and Barbosa (2012) [36], only positive gravity anomalies of interest associated with the mascon are targeted for exploration. Therefore, in this paper, ten seeds (asterisk marker in Figure 13) were employed to invert only target sources (black contours in Figure 14) of the Mare Crisium mascon.

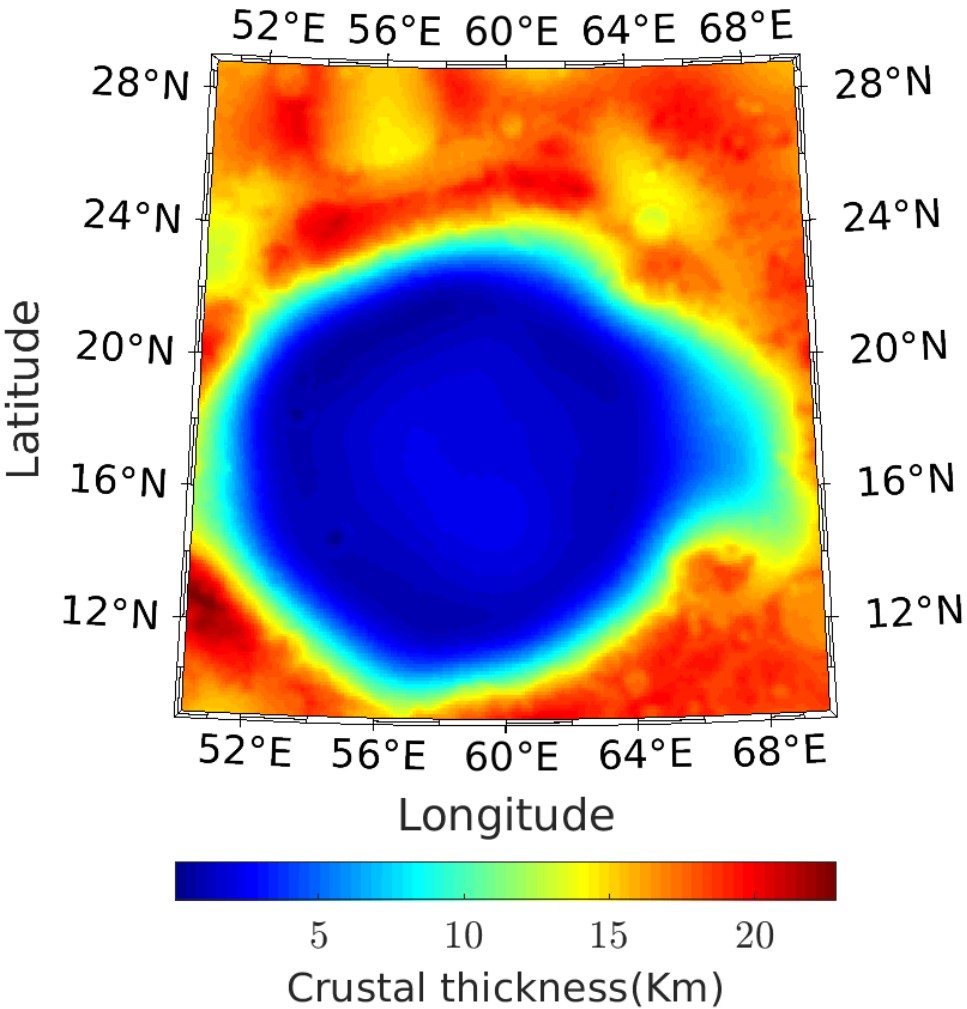

**Figure 12.** The crustal thickness of Mare Crisium.

In spherical planting inversion, seed parameters are generally determined empirically or based on drilling information. Considering that previous studies [40,75,76] used $2.580 \pm 0.170$ g/cm$^3$ as the mean crustal density and $3.22$ g/cm$^3$ as the mean mantle density, and $2.50$ g/cm$^3$ was employed to Bouguer correction for GRGM1200A, the seed density is now set to $0.45$ g/cm$^3$. Because the Mare Crisium mascon is a mantle uplift product, according to the crustal thickness of Mare Crisium in Figure 12, the depth of the seeds is set at 30 km.

For larger or more complex anomalous sources, it is challenging to set individual horizontal weighting functions for each seed, as in Figures 10 and 11 in Uieda and Barbosa (2012) [36], where the recovered density distribution in the right branch is well beyond the limit areas of the horizontal weighting function. Further considering the gravity anomaly characteristics (Figure 9), ten seeds with yellow asterisk markers are now arranged in the positive and negative transition regions of the Bouguer gravity anomaly (Figure 13). Then, the spherical planting inversion is performed using a single seed horizontal weighting function with $\delta = 0.05$ and $\tau = 5.0 \times 10^{-4}$.

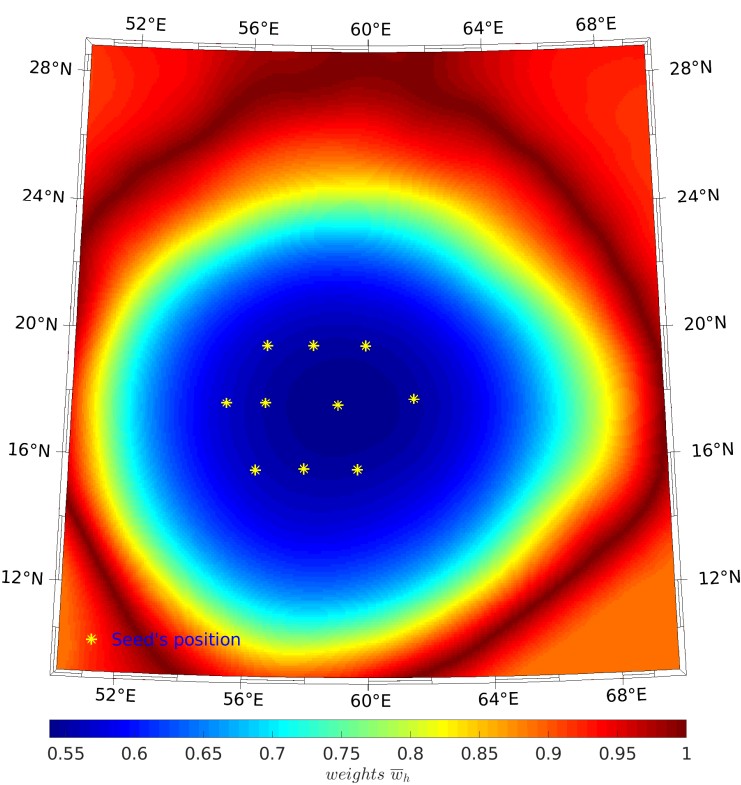

**Figure 13.** New horizontal weighting function applied to field data.

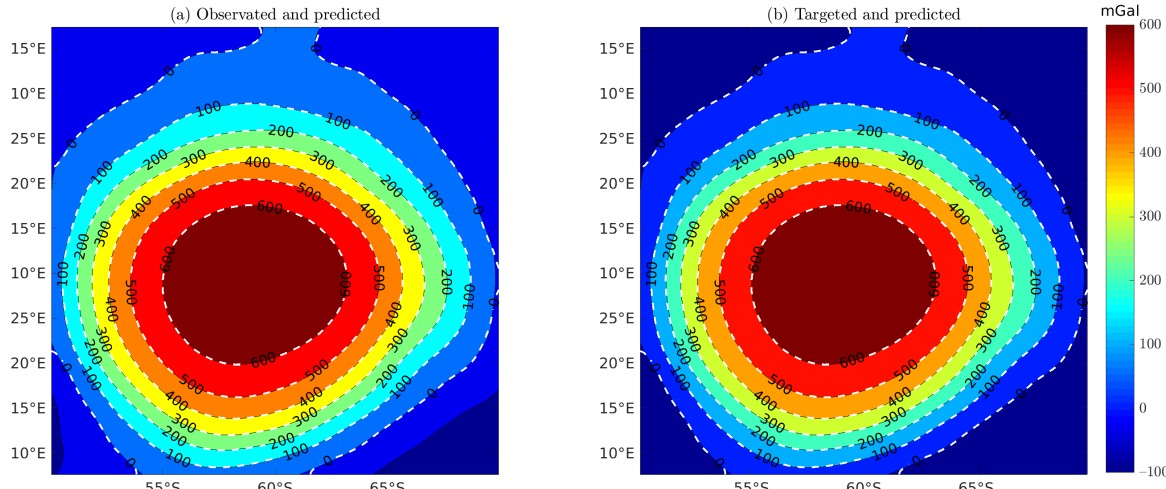

**Figure 14.** Gravity maps. (**a**) Observed data (color−scale map) and predicted data (black contour lines). The observation data is derived from GRGM1200 and is also shown in Figure 9. Due to the low gravity at each corner and the similar linear background field in Figure 9, following Uieda and Barbosa (2012) [36], only positive gravity anomalies of interest associated with the mascon are targeted for exploration. Predicted data is obtained using a spherical planting inversion with ten seeds (Figure 13). (**b**) Targeted data (color-scale map) and predicted Bouguer gravity anomalies (black contour lines).

Compared with Figure 14, the difference between the predicted anomalies of the spherical planting inversion and the exploration target anomalies is tiny, and its norm is 1.2, indicating the present algorithm's stability. Figures 13 and 14 show that the novel seed weighting function fits the predicted data range, indicating that it can effectively constrain the spherical planting inversion [40,77].

## 4. Discussion and Conclusions

Since the tesseroid's length and width vary with its position, we revise the distance calculation formula to avoid preferentially selecting tesseroids closer to their corresponding seeds in the spherical planting inversion.

Because the spherical planting inversion depends heavily on the user's seed parameters, it can incorporate geological priori knowledge to constrain reconstructed density distributions [36,37]. It is like the initial model in iterative inversion. The initial model is not a controlling factor in traditional iterative inversion, and many inversion reports use random initial models. However, the location of seeds in the spherical planting inversion largely determines the spatial spread of the reconstructed density distribution.

Since the gravity field originates from the volume effect of various subsurface anomalous sources [78], incorrect seed densities lead to the recovered density model deviating from the actual one. If the seed density is smaller than the theoretical value, the reconstructed density distribution becomes inflated compared to the actual model. Conversely, the recovered density distribution becomes more compact than the actual one. Because the gravity response rapidly decays with tesseroid depth [79], we found in repeated experiments that, when the threshold is low, the spherical planting inversion tends to select tesseroids near the seeds. As a result, the predicted gravity anomaly obtained from the inversion is similar to the observed one. However, the total mass of the recovered density distribution is significantly larger than the actual model. For this reason, in this paper, we analyze the density/geology studies in this study area and give a more reasonable residual density value for setting seeds.

When a single seed or a few seeds ($N_s < 4$) are used, it was found that the recovered density distribution grew in certain directions and eventually formed a trunk-like/tentacle-like structure, as shown in Figures 10 and 11 in Uieda et al. [36]. In contrast, a compact geological structure is formed in the middle of the recovered density distribution when numerous seeds are used, as shown in Figures 15 and 16, with its top and bottom appearing as scattered tesseroids because there is no tight constraint between seeds according to Equation (4).

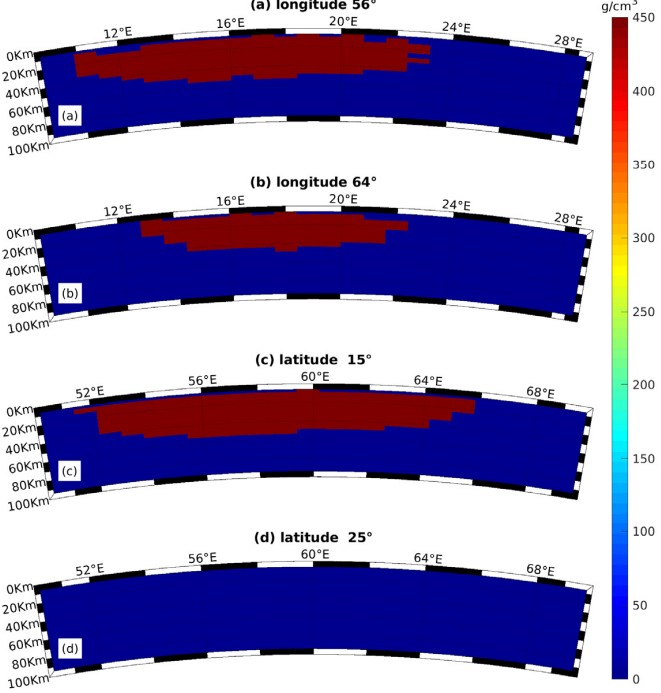

**Figure 15.** Vertical profiles of the recovered density distribution along (**a**) longitude 56°, (**b**) longitude 64°, (**c**) latitude 15°, and (**d**) latitude 25° at Mare Crisium are obtained by spherical planting inversion.

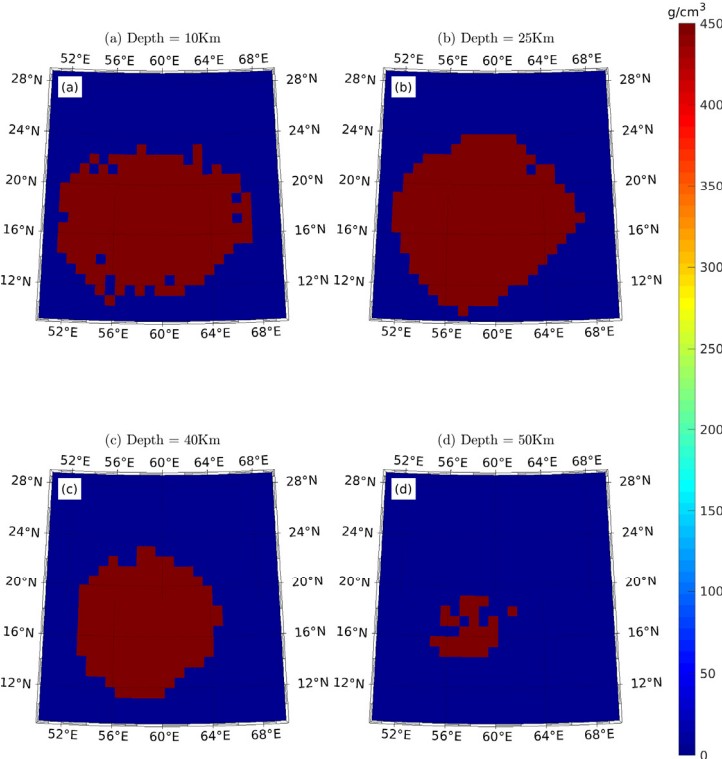

**Figure 16.** Relative density distribution for different depths, (**a**–**d**) are 10 km, 25 km, 40 km, and 50 km, respectively, at Mare Crisium area obtained by spherical planting inversion. Figure 10 shows that the depth of the Mare Crisium basin is about 4–6 km. Figures 15 and 16 show that the mascon has spread from the surface, indicating that the crustal thickness of Mare Crisium is about 0 km because the Crisium impact may have removed all crust from parts of the basin [40,77].

Although tentacles can help optimize seed settings in the subsequent inversion iteration, choosing individual parameters for each seed is difficult, especially when processing large or complex gravity anomalies. For this reason, we propose a novel horizontal weighting function based on $W_d$, as illustrated in Figure 13. Figures 15 and 16 show that the novel horizontal weighting function effectively constructs the subsurface geological structure. It also effectively avoids tentacles in the recovered density distribution.

The algorithm involves calculating the kernel matrix at most once so that the spherical planting inversion of gravity data is faster and has lower memory requirements/consumption than traditional density imaging. The inversion algorithm in this paper is a constrained system search algorithm that achieves fast and efficient spherical planting inversion with few memory requirements. There may be a significant increase in searches in one iteration if the number of seeds is too large.

Correct seed densities are crucial for a spherical planting inversion. Conversely, the recovered density distribution is easily inflated or shrunken compared to the actual geological structure. When the seed's density is smaller than the theoretical value, it is easy to cause the recovered density distribution range to be much more extensive than the actual one. This results in dramatically increasing inversion iterations and reducing inversion efficiency. Therefore, when determining the seed's density, it is necessary to scrutinize the density research history in the study area.

Since the smoothness effect originates from the L2 regulation term [80], the recovered density distributions are smooth everywhere, making it difficult to distinguish the boundaries of geological structures. Also, their density value is far from the theoretical value due to the volume effect of the gravity–magnetic field. There is no depth resolution in surface gravity data. It is challenging to use Occam-like inversion to recover depth

information [21,45,46]. Nevertheless, the spherical planting inversion employed in this paper can preserve depth resolution.

Figures 8–12 illustrate that the depth of the top interface of the Mare Crisium basin is about 4–6 km. Figures 15 and 16 show that the mascon has spread from the surface, showing that the crustal thickness of Mare Crisium is about 0 km because the Crisium impact may have removed all crust from parts of the basin [40,77].

**Author Contributions:** G.L.: Conceptualization, Supervision, Writing—review and editing; D.Z.: Software, Validation, Writing—original draft, review and editing; S.C.: Formal analysis, Methodology, Software, Writing—review and editing; Y.D.: Software, Validation; G.X.: Software, Investigation; Y.L.: Software, Investigation; Z.Z.: Resources; P.C.: Validation, Visualization. All authors have read and agreed to the published version of the manuscript.

**Funding:** This research was funded by the National Natural Science Foundation of China under Grant 41704138, Grant 41974148, and in part by the Hunan Provincial Science and Technology Department of China under Grant 2017JJ3069, and in part by the Project of Doctoral Foundation of Hunan University of Science and Technology under Grant E51651, and in part by the Hunan Provincial Key Laboratory of Share Gas Resource Exploitation under Grant E21722.

**Institutional Review Board Statement:** Not applicable.

**Informed Consent Statement:** Not applicable.

**Data Availability Statement:** Data is unavailable.

**Acknowledgments:** We appreciate the GRAIL mission team distributing the gravity models GL0900C and GRGM1200A. The program in SHTools, which is freely accessible at https://shtools.github.io/SHTOOLS/ (accessed on 4 March 2023), is used to compute gravity anomalies. We appreciate M. Wieczorek for sharing the spherical harmonic model of the shape of Earth's Moon MoonTopo2600p and the software CTplanet for estimating crustal thickness, which is accessible for free at https://github.com/MarkWieczorek/ctplanet (accessed on 4 March 2023). The authors would like to thank D. Gomez-Ortiz and BNP Agarwal for making Parker–Oldenburg's algorithm available. Valuable comments by two anonymous reviewers are gratefully acknowledged. Moreover, we would like to thank Tiffany Liu and the Applied Sciences editorial team for their persistent efforts.

**Conflicts of Interest:** The authors declare no conflict of interest. The funders had no role in the design of the study; in the collection, analyses, or interpretation of data; in the writing of the manuscript, or in the decision to publish the results.

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
