# Peer review of "Spherical Planting Inversion of GRAIL Data"

_applsci, doi:10.3390/app13053332_

Round 1

Reviewer 1 Report

In constructing the kernel matrix, a spherical planting inversion of GRAIL data is proposed using the L1-norm in conjunction with tesseroids. A weighting function was introduced to limit the influence area of the seeds for yielding robust solutions. Moreover, the authors employed the“shape-ofanomaly"datamisfit function in conjunction with a new seed weighting function to improve the spherical planting inversion. The research is innovative. Some opinions are:

1. The reference is not new enough, it is suggested to increase most recent references.

2. Some methods introduction in Section 3 can be adjusted to Section 2.

3. In figure1, it is suggested to give a flow chart for the algorithm.

Author Response

We are very grateful for your comments. I have removed some old, unimportant references, added some relevant ones, and ensured they were added correctly. The section on algorithmic validation in the original draft (subsections 1-2 of Section 3) is moved to subsection 2 of Section 2 of the revised manuscript. A detailed flow chart on planting inversions is provided in the revised manuscript.

Reviewer 2 Report

The authors proposed a spherical planting inversion of GRAIL data, and introduced the weighting function for the robust solutions. The results in the tests show that the proposed inversion is efficient and requires less memory. In addition, it is also demonstrated that the weighting function can limit the area of the seed “growing”. This manuscript is recommended for the publication after the minor revision.

It is hoped that the following comments could improve the manuscript:

1.     In the manuscript, the full name of GRAIL should be given, and the letters of GRAIL must be capitalized.

2.     How to use weighting function in equations (13) and (14)? It should be described in detail.

3.     In Line 181, “i = …”, what is the meaning of i?

4.     In Line 204, “… involves Ns accretion …”, does Ns here mean the number of the seeds? If so, it should be revised as Ns

5.     In Line 255, “… in Equations  and (13), respectively …”, do you mean “Equations (13) and (14)”?

Author Response

We are very grateful for your comments. In light of your comments, we have revised the original manuscript with respect to capitalization, incomplete descriptions of formulae, and a number of other errors.
